# Evaluation of flow, supply, and demand for noise reduction in urban area, Hamadan in Iran

**Shiva Gharibi** [1]*, **Kamran Shayesteh** [2]

**1** Department of Environmental Sciences, University of Kurdistan, Sanandaj, Kurdistan, Iran, **2** Department of Environmental Sciences, Malayer University, Malayer, Hamadan, Iran

☉ These authors contributed equally to this work.

* Shiva.gharibi@uok.ac.ir

**Data Availability Statement:** All relevant data used in this study are available in the supplementary file and available at https://github.com/Gharibi1368/Evaluation-of-flow-supply-and-demand-for-noise-reduction-.git. All the maps (map zip file) were

## Abstract

Noise pollution is one of the consequences of urbanization that can cause environmental disturbances in urban areas. Urban ecosystems provide noise reduction services through Urban Green infrastructures (UGIs). Many studies have been conducted to evaluate and model traffic noise, but none have addressed the flow, supply, and demand of noise reduction ecosystem services. The main purpose of this paper is to present a new methodology for estimating flow, supply, and demand for noise reduction in Hamadan city that has not been mentioned in any paper so far. UGIs were classified into six main categories: agricultural lands, gardens, parks, abandoned lands, single trees, and street trees. A total of 57 sampling stations for sound measurement were made in August 2018. The current map of noise pollution (flow) was created using the Kriging method. The amount of supply was measured up to a distance of 50 meters from the main roads based on two approaches (the distance effect and the sound barrier effect). To quantify the demand, the current sound intensity level in the noise-sensitive land uses was compared with standards. Zonal statistics was used for spatial analysis of supply-demand in the urban neighborhood as a working unit. Results showed that at distances of 5m, 10m, 15m, and 20m, the average noise reduction was found to be 1.61, 2.83, 3.92, and 5.33 dB, respectively. Sound barriers at distances of 5m and 10m resulted in an average sound reduction of 1.61 and 2.83 dB, respectively. Individual trees, strip trees, abandoned lands, parks, and gardens led to a decrease in traffic noise by 0.3, 1, 0.1, 3.5, and 4.5 dB, respectively. The clustering analysis revealed a significant spatial clustering of noise pollution in Hamedan. The results and new methodology of this research can be used in similar areas to estimate the supply and demand of noise reduction and also for decision-makers to take management actions to increase supply and meet the demand for noise reduction service.

## 1. Introduction

Today, over 55% of the population lives in urban areas, which is expected to increase to 60% by 2030 [1]. Noise pollution is one of the consequences of urbanization that can cause

derived using Arc Map by the authors and all data obtained from fieldwork and analyzed (book1 Excel file) have been archived in the Excel file.

**Funding:** The author(s) received no specific funding for this work.

**Competing interests:** The authors also have declared that no competing interests exist.

environmental disturbances [2]. At least, one million lives are lost every year in the western part of Europe due to traffic noise exposure [1]. Due to population growth, the number of people exposed to noise will increase [3], and green spaces will decrease [4, 5] in the coming years. Therefore, the urban environment is composed of several sound sources such as traffic noise [6] which has been characterized as one of the most critical environmental problems in many cities around the world [7, 8] and has been considered the largest producer of noise pollution in urban areas [6, 9–11] that directly affect the life quality [12, 13] after air and water pollution [14]. On the other side, urban green spaces (UGSs), which provide noise reduction services [15], play a vital role in the citizens' quality of life [16]. Road noise pollution can be mitigated both directly (including absorption, deviation, reflection, refraction, and occultation) [17, 18] and indirectly through urban Green spaces [19]. Many studies have been conducted to evaluate and model traffic noise pollution, but none of them addressed the flow, supply, and demand to mitigate noise pollution. The term ecosystem service "flow" refers to the actual and current production or use of ecosystem services (ESs) in a given location [20]. "Supply" is the capacity of a particular region to provide a set of specific goods and services [21], and "demand" is the amount of services required by the individual or desired by society [22].

According to the definitions mentioned above, noise reduction service flow can be defined as the current noise pollution experienced by citizens. Noise reduction service supply is the physical capacity of UGSs to reduce noise pollution [18], and demand is the need for UGSs to compensate for the negative health impacts of traffic noise [23]. Excessive sound in noise-sensitive urban land uses (including educational, medical, and residential centers) annoys residents and employees. Hence, assessing the demand-supply match and mismatch of noise reduction services based on noise-sensitive urban land uses [24] is more significant than other urban land uses and public places.

Reducing the level of sound in educational centers, such as schools, is an important goal of sound studies [25]. Excessive noise can negatively affect students' academic performance and cognitive abilities [26, 27]. So, exposure to traffic noise in educational centers is associated with many problems [28]. Also, noise annoyance in hospitals may cause a serious risk to human health [29] and is the most critical stressor among patients and healthcare personnel [30]. Urban road traffic is the most significant external source of noise in hospitals, which is unavoidable, continuous, and increasing. Therefore, it is important to pay close attention to the demand for traffic noise reduction in the patient's recovery process [31]. Residential areas are the other sensitive land use that can adversely affect the physiological and psychological well-being of citizens. Still, today they are exposed to major problems, especially excessive noise pollution [32].

Finally, an assessment of the match and mismatch between the supply of and demand for noise reduction ecosystem service in noise-sensitive land uses including hospitals, schools, and residential can be an appropriate solution to increase human well-being in urban areas. Due to the lack of noise studies in quantifying the supply of and demand for noise reduction ecosystem service and the long-term consequences of traffic noise pollution on individuals, especially in large cities, the importance of the matter increased. Also, quantifying and mapping ecosystem services supply provides information about the status and accessibility of production areas. Demand also considers stakeholders' distribution [33]; No such studies have been performed for a noise reduction service. On the other hand, the demand for traffic noise reduction has been ignored because of its natural and social complexity. Therefore, the methodology of evaluating and quantifying the supply, demand, and flow of traffic noise and its relationship with urban green infrastructure (UGI) is essential and has been addressed in this study. Therefore, this study aims to quantify and map the supply-demand ratio of noise reduction in the Hamadan urban area and shows the role of UGLs in providing the reduced

and demanded noise reduction ecosystem service. Also, it is assumed that there is a mismatch between supply and demand in Hamedan city.

## 2. Methodology

First, the Hamadan Comprehensive Plan was used to extract the urban road network. Then, the main streets, highways, and boulevards were chosen as the principal sections in terms of noise pollution (high speed and high congestion). The urban land use map (in 2018) was obtained from the Hamadan Municipality Organization as well. To map the UGIs and investigate the association between green spaces and traffic noise, Lopez et al. [34] methodology was employed. The classes of strip trees (any tree growing in the City's right-of-way), single trees, gardens (a dense group of trees), parks (land covered with sparse trees and lawns), Agricultural lands (an area under crops), and abandoned lands (or grass cover, are those once used to grow crops or as pasture, but no longer used) were taken into account. Using images from Google Earth, all green space polygons of various sizes and forms were distinguished and recognized. Terra Incognita was used to count the pixels of images, and ArcGIS was used to analyze the result and determine the percentage and kind of green space in the study area. Besides the classification algorithm, field observations were used to validate the results [34].

### 2.1 Study area and data sources

Hamedan is one of the most important cities in Iran and the capital of Hamedan province. Its population in the 2019 census was slightly over 554,000. According to the Statistics and Urban Green Space Organization of Hamadan, the city has an area of about 5627 ha. This city is located at the foot of Alvand Mountain at an altitude of 1900 meters above sea level and is one of the coldest cities in Iran. Fig (1) shows the location map of the study area. Noise reduction, as one of the regulating ecosystem services, was also selected as the most critical service

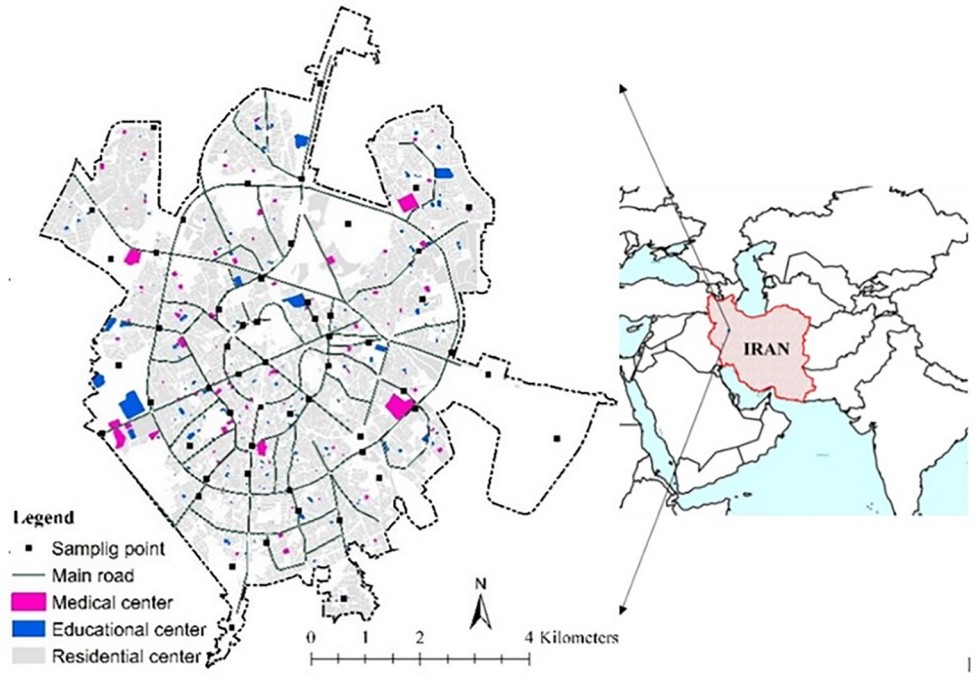

**Fig 1. Location map of Iran, Hamedan province, and the urban land uses studied in Hamedan city.**

required for human welfare. Since Hamedan frequently experiences noise pollution, a concern that has gotten less attention than other issues. Although issues like traffic have contributed to the noise situation, Hamedan Urban Management's administrative and management authorities have not considered it. The urban area is divided into 75 urban neighborhoods which were selected as working units.

## 2.2 Sound measurement

Whereas this study aims to investigate the reduction of noise pollution by UGIs, and the evergreen trees are particularly important in this regard [35], sound measurements were made in August, when traffic and vegetation were at their highest. A total of 57 sampling stations were chosen through the study area. First, 25 of those were selected based on the $NO_2$ and CO pollution maps obtained from the Hamedan urban area's Pollution Management Plan, in 2018 (these points were selected in the air pollution hot spots). Air pollution and noise pollution have a spatial correlation, as reported by Bloemsma et al. [5]. The other 50 points were chosen randomly and scattered throughout the city. Then, the sound level was recorded using CEL-450, with a measurement range of 40–120 and a precision of 0.1 dB. The sound device was placed at a distance of one meter from the roadside [36] and 1.2 meters from the ground [37] in the middle of the road [38]. Leq (A) was constantly recorded for 15 minutes at each location [37]. The microphone was covered with a sponge protector to mitigate the effect of wind [39]. The measurements were adjusted to exclude the additional natural and human noises. All measurements were performed under the same conditions on rainless days with minimal wind and over two weeks as a representative of long-term sound changes [40] from Saturday to Wednesday from 7:00–9:00, 13:00–15:00, and 18:00–20:00 in September. Existing noise status is a function of sound production, and the noise reduction rate is affected by various parameters. Therefore, regardless of the factors affecting sound production only the existing traffic noise level was measured.

## 2.3 Quantification and mapping of the flow

The flow refers to the current sound level in the Hamadan urban area. The current map of noise pollution was created by applying the Kriging method as a flow of traffic noise level in the study area and categorized into nine classes to determine noise pollution hotspots (values above the standard) using a geographic information system (GIS). Kriging is a method of interpolation based on the Gaussian process which gives the best linear unbiased prediction at unsampled locations [41].

## 2.4 Quantification and mapping of the supply

Supply refers to the capacity of UGIs to provide noise reduction as an ecosystem service. Since most noise reduction happens up to a distance of 50 meters from the main roads (highway, main street, and boulevard) only because of UGIs, it is considered as noise reduction supply, and beyond that, sound waves are blocked by the buildings [18]. According to Fig 2, sound measurement was performed simultaneously at the road's edge (station A) based on the different distances (first approach) and the same distances (second approach) from the road (station B). For the physical or vegetative barriers between stations A and B, all data was recorded. The noise reduction supply can be mapped based on two approaches. The study by Ow and Ghosh [42] was utilized to determine the effect of distance and plant/non-plant barriers on noise reduction. The study by Ow and Ghosh [42] was used to determine the effect of distance and plant /non-plant barriers on noise reduction.

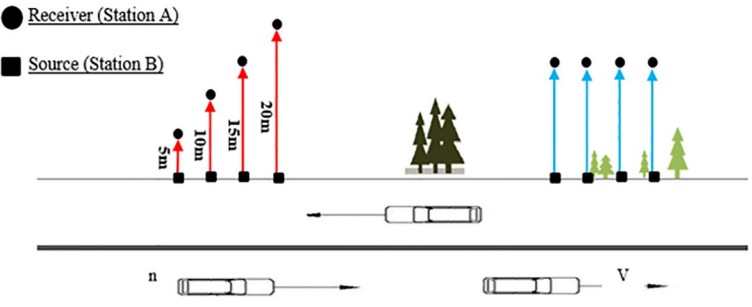

**Fig 2. Sampling plan of measuring noise reduction by distance and sound barrier (two approaches).**

In the first approach, there are no barriers between the source (station A) and the receiver (station B). Consequently, a linear fuzzy membership function (monotonically increasing) is used to remove the distance effect. Therefore, the first approach involved recording equivalent sound level (Leq) simultaneously for up to 15 minutes at each of the four stations, with nine repetitions per station. At the edge of the road (station B, noise source) and the distance of 5, 10, 15, and 20 meters (station A, receiver), without any noise barriers between the noise source (traffic) and the receiver (sound meter), Leq was recorded to assess the effect of distance on sound reduction in Fig 2 (red arrows).

In the second approach, the quantity of decreased sound was considered as the sound barrier effect by measuring the sound level at locations A and B and then removing the distance effect. Therefore, the second approach involved taking simultaneous noise measurements at two stations, A and B, at fixed distances of 5 and 10 meters to assess the effect of sound barriers (plant and non-plant). Next, the amount of dB reduction was obtained due to the various kinds of sound barriers using Eq (1).

$$BE = NR - DE_{5:10} \tag{1}$$

Where; BE is the reduced sound by acoustic barriers (dB), NR is reduced sound (difference in sound level measured at points A and B simultaneously) in dB, and DE is the reduced sound by distance (5 and 10 meters) in dB. Fig 2, shows sound recorded in two measuring points A and B based on two approaches.

To calculate noise reduction supply by the UGIs, all of the information, including the distance and the type of noise barrier between the noise source (traffic) and the receiver (sound meter), was recorded. Afterward, the results of the second approach measurement were applied to the sound flow map, and the Leq map was created without taking into account the effect of the sound barrier. The value of the roadside green spaces (in the 50-meter buffer) acts as a noise reduction supply. So, all existing UGIs (including gardens, abandoned lands, single trees, strip trees, and agricultural lands) in the 50-m buffer were extracted and evaluated based on their amount of noise reduction effect which was estimated based on the second approach. Finally, the share of each neighborhood in the noise reduction supply was done using zonal statistics with GIS.

## 2.5 Quantification and mapping of the demand

Demand refers to the amount of noise reduction required by noise-sensitive land uses. Noise-sensitive land uses, including medical centers (hospitals, clinics, and emergencies), educational centers (primary schools, high schools, universities, and higher education centers), and

**Table 1. Iranian national noise standards in sensitive land use.**

| Landuse | | Day (7 am—10 pm) | Night (7 pm—10 am) |
|---|---|---|---|
| Residential area | | 55 | 45 |
| Educational Centers | | 45 | - |
| Medical centers | Open space around a hospital | 55 | 45 |
| | Free space inside a hospital | 45 | 35 |

residential areas, were determined based on the comprehensive plan of Hamadan, field investigation, and the use of Google Earth satellite images. To quantify the demand, the current sound intensity level in the sensitive land uses was retrieved using the sound flow map, and its comparison with the Iranian national noise standards (Table 1) can indicate the degree of demand in each use [43]. Sound mapping was carried out independently of interior noise sources and noise reduction by walls, windows, and other sound barriers was ignored.

Finally, Spatial Analysis is performed in the neighborhoods of Hamedan as working units using the Zonal Statistics tool (mean statistics). Sound level and green spaces are extracted for each working unit, and the G statistic is used to check the presence or absence of clustering. Getis-Ard G (statistic Gi) is used to identify hot and cold spots in the Arc map. To determine the distribution of noise pollution-green space patterns, the Moran index is used to examine their spatial autocorrelation. Then, the min and max distances obtained from the zonal statistics were reclassified using Jenks statistical processes.

The supply-demand ratio analysis is performed in two steps. (1) The supply and demand for each working unit are mapped to visual observation of the difference between supply and demand. (2) The supply-demand ratio for each working unit is estimated to determine the ability of supply to meet demand [44]. Eq (2) is used to calculate the supply-demand ratio. Where; R is the supply-demand ratio, S is the supply, and D is the demand for noise reduction. The resulting values are standardized in the range of 0–100.

$$R = {^S/_D} * 100\% \tag{2}$$

## 3. Results and discussion

According to the results, the residential area, UGIs, educational, and medical centers occupy 3149, 1239, 89, and 87 ha of the urban area, respectively. The majority of the city is made up of road networks and constructed environments, with only 22% being UGIs. GIs were classified into six main categories: agricultural lands, gardens, parks, abandoned lands, single trees, and street trees. Each class accounts for 37.26, 10.58, 21.57, 17.34, and 13.26 (single and street trees) percent of UGIs, respectively. Urban trees cover 2.92% of the total study area. There are approximately 1126884 trees. Hamedan trees were mostly young (70% of individual trees were less than 23cm in diameter) and comprised a variety of species, including Elm, Black locust, Maple, and Manna Ash. To validate the UGI classes, 24 points were picked on the map, and the type of green space was recorded by fieldwork. Based on the results, 22 sampling points (92% validity) corresponded to the infrastructure classes defined.

### 3.1 Traffic noise flow

The noise flow in Hamedan is depicted in a color spectrum from green to red (63.8–91.7) on a scale shown on the right side of Fig 3. The red areas have the highest average equivalent sound level (Leq), while moving towards the green areas reduces the intensity of noise. The highest and lowest Leq are associated with station 12 (Farhangian Town bridge) at 99 dB, and station

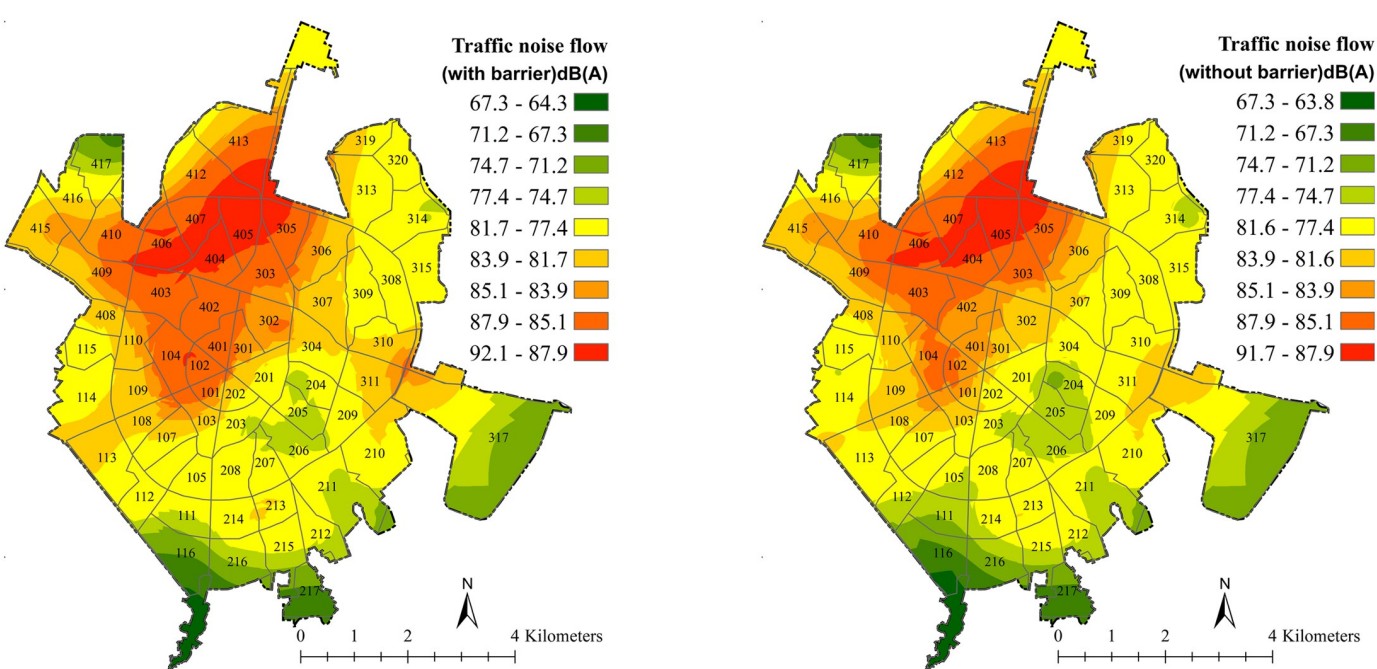

**Fig 3. Traffic noise flow and locations of hotspots with considering (left); and without considering the sound barrier effect (right) based on the Kriging method.**

48 at 55 dB, respectively. Noise mapping in the Hamedan urban area, with an area of 56.27 km$^2$, was done involving sound measurement in 57 sampling stations. Lan et al. [37] also performed sound mapping and modeling in the Chancheng district with an area of 150 km$^2$ in 30 sound stations. The validation results showed the high accuracy of the model [45] and Bloemsma et al. [5] used the NDVI index to determine the effect of green spaces around residential homes, which is less precise than identifying tiny polygons and single trees. According to the flow map, the sound level varies between 63.8 to 91.8 dB. According to Katorani [46], the average sound level in Hamedan city is 80.4 dB, whereas it is 72 dB in Sanandaj city. More people are moving to cities as the rate of urbanization rises, resulting in less green space and more air pollution, including noise pollution [5]. Station 12 (Farhangian town bridge) has the highest recorded sound pressure level. This station is on the main line, near the Hamedan International Exhibition, and adjacent to three residential communities. The industrial facility, which is near the bus passenger terminals, emits excessive noise. The lowest sound level (station 48) is recorded inside the agricultural land, distant from the main road. The recorded sound level in São Carlos city is 56.18–71.39 at night and 55.64–76.45 during the day, which is an alert range for a medium-sized city [34], but the sound level in Hamedan city is critical.

## 3.2 Supply of noise reduction service

The distance effect on noise reduction revealed that as the distance from the roadside increased to distances of 5m, 10m, 15m, and 20m, the average noise reduction was found to be 1.61, 2.83, 3.92, and 5.33 dB, respectively (Distance without any sound barrier and with the ground surface of concrete and paving). According to Ow and Ghosh's results [42], the amount of sound reduction at intervals of 5, 10, and 20 meters is 1, 3, and 4 dB, respectively. It was carried out in conditions with little greenery (grass cover) and no sound barriers. According to Rochat's research [47], doubling the distance from the sound source reduces the sound

by 3.5–5 dB. However, in the present study, the decrease is slightly lower at 2.50–2.83 dB. This may be due to the specific characteristics of the urban area in Hamedan and the presence of various obstacles that affect sound propagation. According to Rochat [47], doubling the distance from the noise source reduces road traffic noise by 3.5–4 dB. Without a sound barrier, the average noise reduction effect due to the distance from the sound source (5–20 m) is 1.5–5 dB. Sound is attenuated by increasing the distance between the sound source and receiver due to friction between atmospheric molecules as the sound moves [48]. The shorter the distance, the more effective [49].

Noise reduction can also be affected by elements other than vegetation, such as temperature and humidity [50]. As a result, measurements were carried out under the same weather circumstances to assess the effectiveness of a sound barrier in lowering road noise. Additionally, the presence of sound barriers at distances of 5m and 10m resulted in an average sound reduction of 1.61 and 2.83 dB, respectively. Different types of sound barriers also resulted in different average sound reductions; individual trees, strip trees, abandoned lands, parks, and gardens led to a decrease of traffic noise by 0.3, 1, 0.1, 3.5, and 4.5 dB, respectively. GIs reduce noise pollution, similar to the findings of Jaafari et al. [51] and Alikhani et al. [52] Density, width, height, length of tree belts, leaf size, and tree branching characteristics are all important vegetation factors for noise reduction [18]. Noise is reduced by 0.3 dB when single trees are used as sound barriers. According to the research of Derkzen et al. [19], individual trees have a small and zero reduction effect in reducing traffic noise. Hamedan's trees are young and comprised of many species such as Black locust, Ulmus, Maple, and Manna Ash. Various plant species mitigate noise differently [53]. According to Makhdoom [54], Oak, Sycamore, Black locust, and pine species have a more sound-reducing effect; However, strip trees (average canopy width of 7 m$^2$) as a sound barrier reduce the sound by 1 dB. Sound waves also pass through hard surfaces such as asphalt, cement, and stone faster, and soft surfaces such as grass or other vegetation can reduce noise more. Therefore, planting trees and lawns near noisy places reduces noise significantly. In this study, grass cover as a sound barrier reduces sound level by 0.1 dB. However, according to the study of Derkzen et al. [19], this level of decrease was reported as 0.4 dB. Sound levels are reduced in parks and vegetated areas with tall Sycamore trees [46]. In this study, the reducing effect of parks as sound barriers (with an average width of 15m) resulted in a traffic noise reduction of 2.9 dB. A bunch of trees (gardens) also reduce the noise received by 4.5 dB (average width of 20 m). Similarly, a 30-meter-wide cropland with tall and thick trees can reduce noise by up to 50% (equivalent to a reduction of 10 dB or more [55]. According to Derkzen et al. [19] research, the number of garden spots in the 50-meter buffer was rare since the majority of garden spots are further away from the city center and main roads, and the amount of noise reduced by the garden is considered zero. The sound level within the 50-meter buffer zone of main roads varies from 64.91 to 91.76 dB. To determine the noise reduction supply the distance between the noise sources and receivers as well as the types of sound barriers were recorded. Then, based on the first approach (distance effect), the second approach (sound barrier effect) was estimated and applied to the sound flow map to produce the equivalent sound level map without considering the effect of sound barriers (Fig 3, left).

To prepare a map showing the supply of noise reduction by UGIs, first, the value of each type of GI in reducing traffic noise within the 50-meter buffer zone was determined (multiplying the area of each GI by the value of noise reduction). then, the contribution of each neighborhood in reducing noise pollution was extracted based on the zonal statistics (The average noise reduction for the gardens, parks, strip trees, single trees, and grass cover are 4.5, 3.3, 1, 0.3, and 0.1 dB, respectively); see (Fig 4). According to the results, the amount of noise reduction due to green sound barriers ranges from 0 to 4.5 dB. On average, gardens, parks, strip

trees, single trees, and grass cover were able to reduce traffic noise levels by 4.5, 3.3, 1, 0.3, and 0.1, respectively. On average, the mean of noise reduction by green barriers is 0.1–6.4 dB which is 9–11 dB in the study of Ow and Ghosh [42]. The spatial analysis result showed that the highest supply of noise reduction is in neighborhoods 119 and 316, where the area of green spaces (in the 50-meter buffer) are 30926 and 2309 $m^2$, respectively. With 5807 $m^2$ of green space, neighborhood 411 has the lowest supply. As a consequence of Kia's study [32], residential areas presently deal with countless problems, especially noise pollution; In this way, the sound level in the Hamedan residential area (45–91 dB) exceeded the standard. The most demand in residential areas is in the neighborhoods 405, 406, 407, 404, 305, and 403, respectively. When compared to other types of uses, such as residential, commercial, and so on, educational centers, which are one of the most critical locations, must adhere to strict sound levels [25]. According to Noweir and Ikhwan [56], sound pressure levels are higher in overcrowded urban areas and schools close to the street. The sound level in Hamadan educational centers is between 64.41 and 89.63 dB. The neighborhoods with the highest demand are 415, 413, 407, 406, and 405, respectively. Golmohammadi and Aliabadi [31] showed that the main external source in medical centers is traffic noise, which is unavoidable, continuous, and increasing. In this study, the sound level varies between 69.65 and 90.79 dB; which indicates that the studied hospitals suffer from noise pollution.

## 3.3 Demand for noise reduction service

Because the sound measurement was done during the peak time of urban road traffic, only a comparison of demand with the sound standard was made during the daytime for the three land uses. The amount of noise in the residential areas varies from 91 to 45 dB. Compared to the standard (55dB) most areas are exposed to high noise pollution. The sound level at educational centers varies from 64.41 to 89.63 and is high compared to the standard (45 dB). Also, the highest demand for noise pollution reduction is in 405, 406, 407, 413, and 415 neighborhoods. The sound level at health centers varies between 69.65 and 90.79 dB. So, the sound level is higher than the standard level (35–55 dB) in all medical centers. A percentage of this noise pollution is reduced by walls, windows, distance from the road, and the presence of a yard which has been neglected. Fig 5 shows the final map of the demand level in the three residential, educational, and medical sectors. Sound levels above 80 dB are noise pollution, and reducing them to standard levels is considered as demand. Based on this, neighborhoods 101,104,107, 110, 113, 114, 201, 202, 213, 214, 302, 307, 308, 311, 319, 402, 404, 410, 412, 413, and 415 have the highest demand for reducing noise pollution (Fig 5). Table 2 also shows the average sound level in each neighborhood of the Hamadan urban area.

Based on the spatial analysis, the most demand for noise reduction in medical centers is related to 405 and 407 neighborhoods, followed by 406, 404, 413, 415, 412, 102, 104, and 410. The lowest demand is related to 118 and 411 neighborhoods. The lowest sound level is related to neighborhoods 119 and 117 and the highest sound level is related to 405, 406, and 407. In terms of the importance of green space in a 50-meter roadside buffer in reducing noise pollution, neighborhoods 418, 109, and 309 have the lowest importance. The study also compared its findings to previous research, showing that increasing the density of green barriers can further decrease traffic noise. The spatial analysis revealed that certain neighborhoods had higher demand for noise reduction, particularly in residential, educational, and medical areas. Despite neighborhood 407 having a relatively high ratio of green space to its area, it still had a high demand for noise reduction. Neighborhoods 116 and 117 were identified as having the highest importance and value in reducing noise pollution, while neighborhoods 405, 406, and 407 had the highest demand for noise reduction across all three residential, educational, and medical

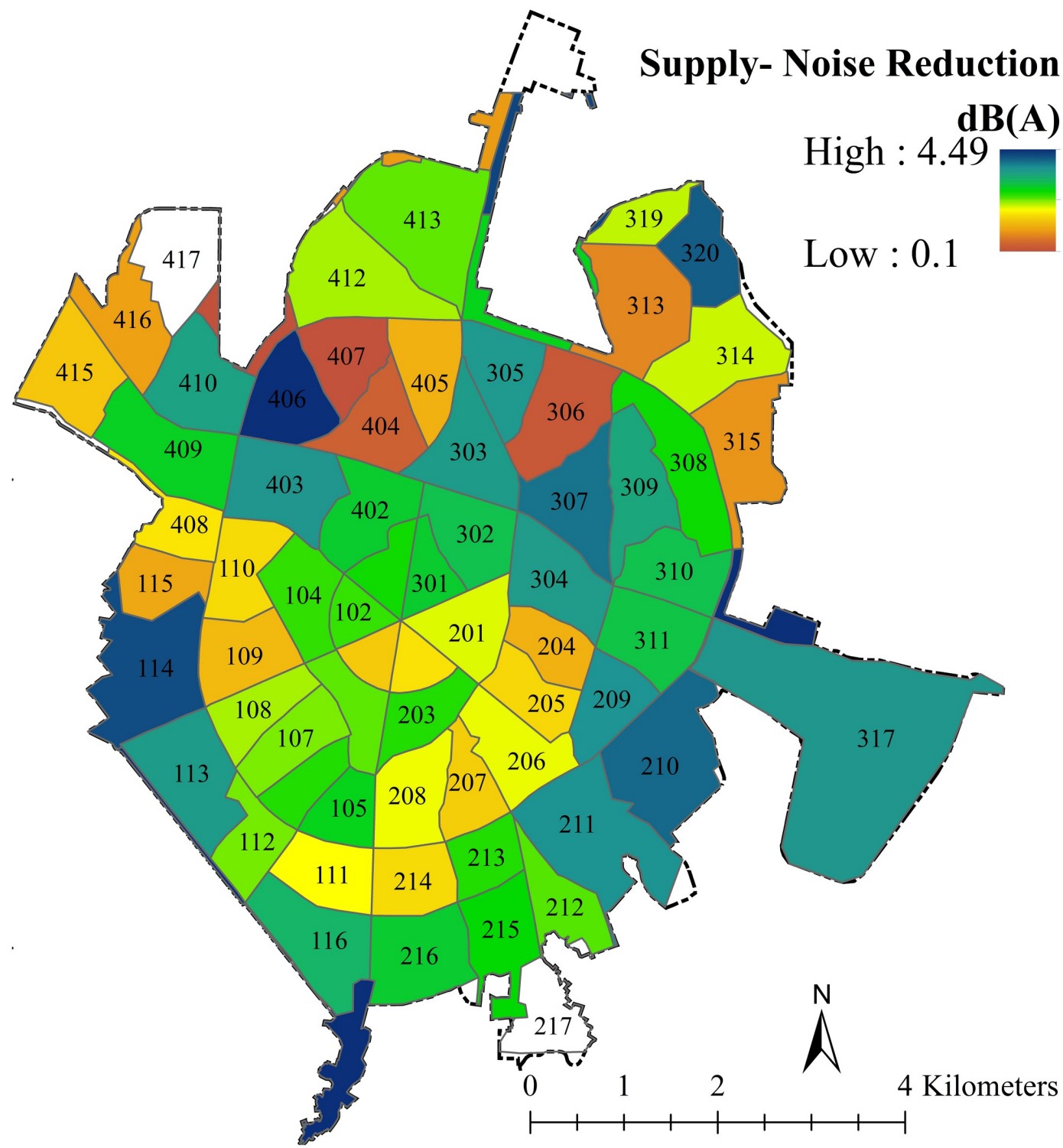

**Fig 4. The map of noise reduction supply by UGIs in each neighborhood using the zonal statistic tool.**

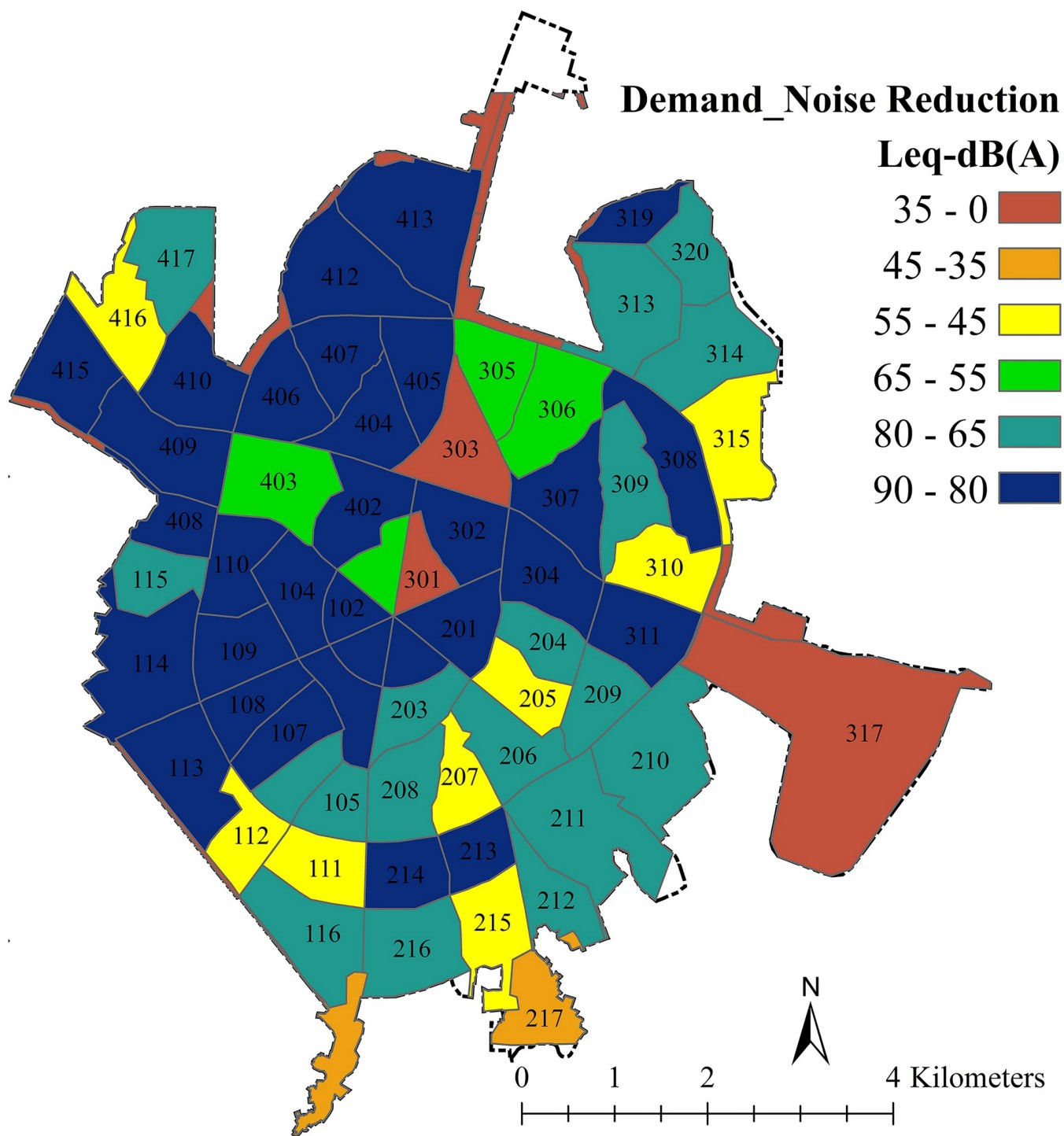

**Fig 5. The demand map for noise reduction in each neighborhood using the zonal statistic tool.**

uses. Despite neighborhood 407 having a relatively high ratio of green space to its area (1.05%), it still had a high demand for noise reduction.

There are 400 trees, 33.4 ha of gardens, 32.5 ha of agricultural lands, 199 m$^2$ of parks, 5.5 ha of abandoned lands, and 400 trees (in total 71.56 ha) in neighborhood 407. Within the

**Table 2. Average sound level in each neighborhood of the Hamadan urban area.**

| Code | Leq(dB(A)) | Code | Leq(dB(A)) | Code | Leq(dB(A)) | Code | Leq(dB(A)) |
|------|-----------|------|-----------|------|-----------|------|-----------|
| 101 | 84.06 | 201 | 80.87 | 301 | 27.39 | 401 | 56.23 |
| 102 | 85.41 | 202 | 81.20 | 302 | 82.34 | 402 | 84.92 |
| 103 | 82.16 | 203 | 77.39 | 303 | 28.33 | 403 | 57.13 |
| 104 | 85.14 | 204 | 70.87 | 304 | 80.23 | 404 | 87.63 |
| 105 | 77.33 | 205 | 50.82 | 305 | 57.58 | 405 | 89.26 |
| 106 | 79.27 | 206 | 77.07 | 306 | 56.25 | 406 | 88.22 |
| 107 | 80.83 | 207 | 52.50 | 307 | 81.27 | 407 | 88.97 |
| 108 | 82.93 | 208 | 77.96 | 308 | 81.07 | 408 | 82.05 |
| 109 | 83.39 | 209 | 78.39 | 309 | 79.09 | 409 | 84.86 |
| 110 | 83.31 | 210 | 79.63 | 310 | 52.56 | 410 | 85.06 |
| 111 | 50.71 | 211 | 78.18 | 311 | 81.13 | 411 | 26.73 |
| 112 | 51.76 | 212 | 78.24 | 312 | 28.14 | 412 | 85.44 |
| 113 | 81.79 | 213 | 82.35 | 313 | 79.55 | 413 | 85.63 |
| 114 | 80.55 | 214 | 80.42 | 314 | 77.97 | 414 | 0.00 |
| 115 | 79.51 | 215 | 53.57 | 315 | 52.63 | 415 | 85.47 |
| 116 | 70.74 | 216 | 74.99 | 316 | 27.68 | 416 | 54.06 |
| 117 | 0.00 | 217 | 44.65 | 317 | 0.00 | 417 | 72.68 |
| 118 | 25.22 | | | 318 | 27.97 | 418 | 0.00 |
| 119 | 43.22 | | | 319 | 81.08 | 419 | 27.70 |
| | | | | 320 | 79.05 | | |

50-meter buffer, however, there are only 3.2 ha of GIs, of which 2.8 ha are agricultural lands which have the lowest value in reducing noise pollution. There are 27.23 ha of GIs in neighborhood 405- a ratio of 0.37 to the neighborhood's area. Agricultural lands cover 18 ha, gardens cover 4.6 ha, parks cover 2 ha, abandoned lands cover 2.4 ha, and a canopy of 299 trees cover 0.37 ha. However, there are 4.9 hectares of green infrastructure within the 50-meter buffer, and there are 4.9 ha of GIs, including 3.3 ha of agricultural lands, 1.1 ha of parks, 0.3 ha of gardens, and 0.2 ha of tree canopy. Neighborhood 406 also covers 22.2 ha of GIs, a ratio of 0.28%, including 9.1 ha of agricultural lands, 1.2 ha of gardens, 0.86 ha of parks, 11 ha of abandoned lands, and 539 m$^2$ of trees.

Therefore, the presence of garden uses with the greatest potential for noise reduction in comparison to other green uses and the low population living is the best reason for the low noise level and the decrease in demand for noise reduction. Based on the spatial analysis and clustering of noise pollution in the Hamedan urban area, the most affected neighborhoods by noise pollution are 404, 405, 406, 407, 412, and 413. There is a 99% probability that they are hot spots of noise pollution. In conclusion, noise pollution is a significant issue in urban areas, and it is influenced by factors such as urbanization, traffic density, and distance from sound sources. The study also showed that increasing the distance from sound sources can effectively reduce noise levels. Therefore, urban planning and design should consider measures to mitigate noise pollution and improve the overall quality of life for residents by increasing GIs.

The results of zonal statistics are shown in Fig 6. Each pixel value (neighborhood) indicates the average density of the parameters under study. Red colors indicate the highest share of UGIs (Fig 6, right) and noise pollution (Fig 6, left), with a shift towards yellow indicating a decrease in both parameters on both maps. To determine the zonal statistic analysis of green spaces, the abandoned land category was excluded from the analysis as they did not affect noise reduction.

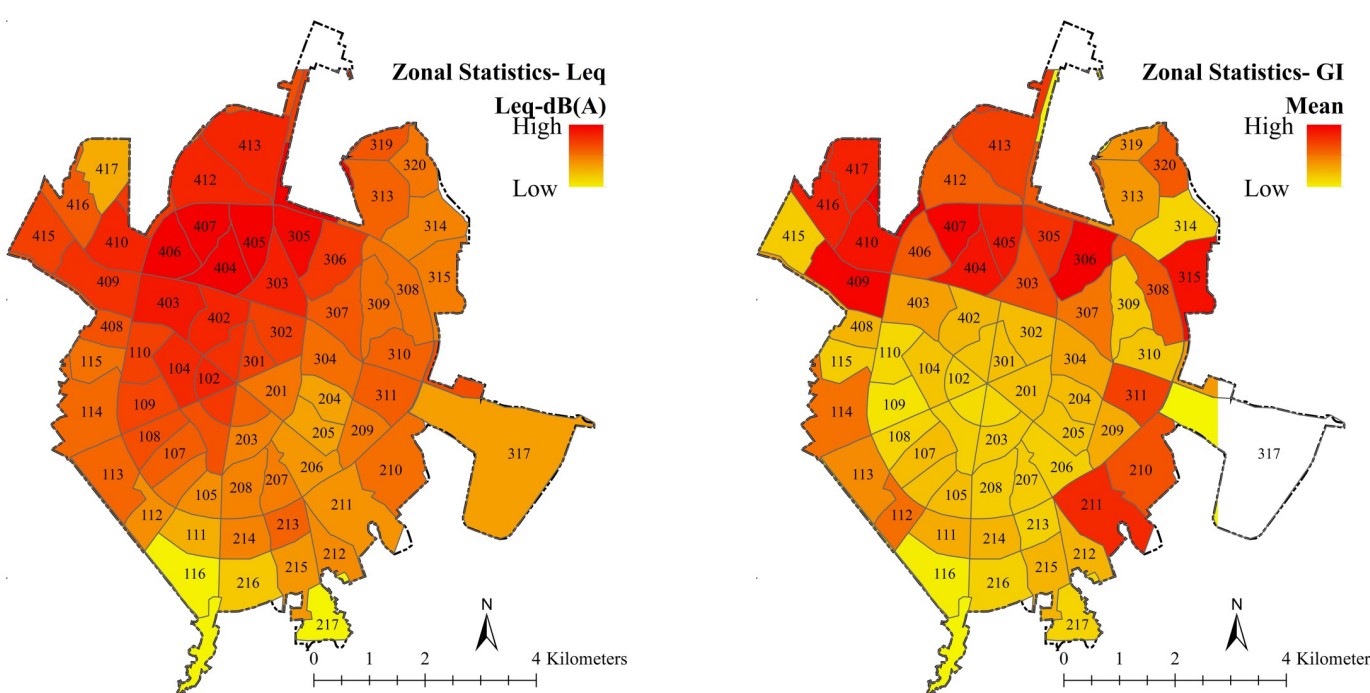

**Fig 6. Zonal statistic analysis of noise pollution (left) and UGIs (right) based on the average statistics.**

The clustering analysis revealed a significant spatial clustering of noise pollution in Hamedan. The G statistic rejected the null hypothesis of no spatial clustering, indicating that high levels of noise pollution are clustered in the area (Z value). Moran's statistic showed a positive value close to 1 (0.76), indicating spatial autocorrelation and cluster distribution pattern of noise pollution. The area analysis depicted in Fig 7 showed that areas with red colors have high noise levels and low importance of GIs in reducing noise pollution. As the color moves towards yellow, the noise pollution decreases and the importance of GIs increases. This indicates that there is heterogeneity in the supply and demand for noise pollution reduction ecosystem services. Table 3 presents the results of spatial analysis of noise pollution and the importance of GIs, showing their comparable relationship in the range of 0–100. This analysis provides valuable insights into the distribution and impact of noise pollution and GIs in Hamedan.

Fig 8 shows the supply-to-demand ratio to reduce traffic noise. The color spectrum reveals that dark blue and green have the highest supply-to-demand ratio, but decreases when the color is switched to yellow and purple. The demand is the average sound level difference from the threshold (80 dB). The values 46 mean supply and demand are equal. Values over 46 mean more supply than demand, and the larger the value, the much more supply than demand. Values 1–46 mean demand is more than supply; the smaller the value, the greater the demand. Values between 0–41 mean no demand.

## 4. Conclusion

Road distance and plant/non-plant barriers have the main effects on noise reduction. To measure these two important factors, two different approaches have been considered (the distance effect and the sound barrier effect). Without a sound barrier, the average noise reduction effect due to the distance from the sound source (5–20 m) is 1.5–5 dB. The

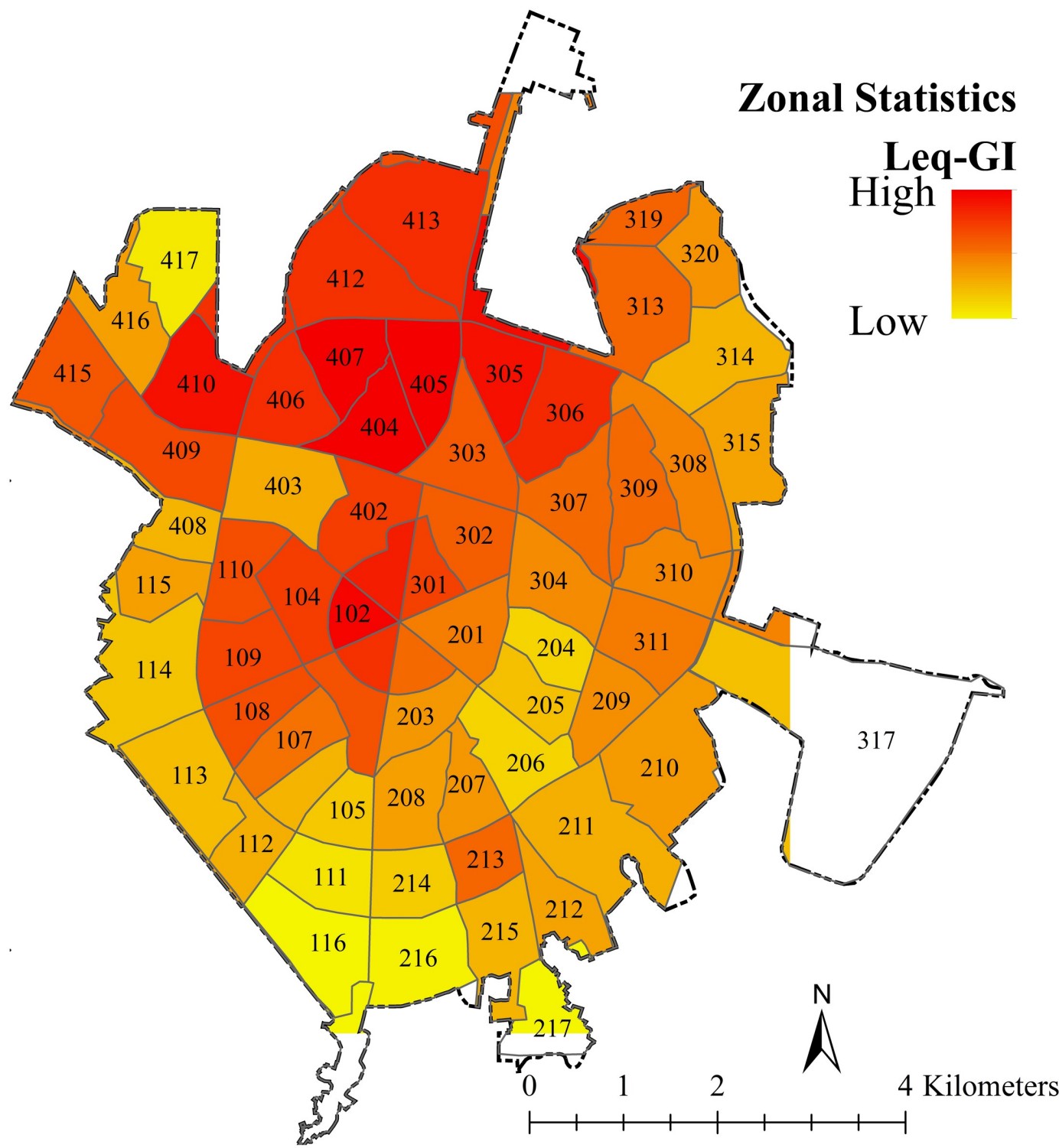

**Fig 7. Noise zonal statistics—The importance of GI in reducing noise pollution in urban neighborhoods.**

**Table 3. Results of zonal statistics of noise (dB) and the importance of GIs in reducing noise pollution (0–100).**

| Code | Leq | GI | Code | Leq | GI | Code | Leq | GI | Code | Leq | GI |
|------|------|-------|------|------|-------|------|------|-------|------|------|-------|
| 101 | 83.79 | 1.98 | 201 | 79.21 | 3.47 | 301 | 82.96 | 2.11 | 401 | 84.35 | 1.98 |
| 102 | 99.89 | 0.31 | 202 | 80.54 | 2.24 | 302 | 83.09 | 1.76 | 402 | 85.10 | 5.76 |
| 103 | 81.91 | 1.91 | 203 | 78.25 | 2.27 | 303 | 85.43 | 6.32 | 403 | 86.15 | 43.17 |
| 104 | 84.93 | 3.79 | 204 | 76.12 | 2.17 | 304 | 79.80 | 3.45 | 404 | 87.83 | 1.11 |
| 105 | 77.51 | 6.16 | 205 | 76.06 | 1.19 | 305 | 87.28 | 2.34 | 405 | 89.29 | 1.46 |
| 106 | 79.11 | 6.22 | 206 | 76.77 | 13.17 | 306 | 83.62 | 0.30 | 406 | 88.02 | 13.76 |
| 107 | 81.04 | 2.03 | 207 | 78.62 | 1.03 | 307 | 81.42 | 2.39 | 407 | 88 | 1.51 |
| 108 | 82.30 | 0.82 | 208 | 7840 | 1.97 | 308 | 79.69 | 2.09 | 408 | 81.98 | 26.46 |
| 109 | 82.71 | 0.01 | 209 | 78.82 | 2.57 | 309 | 79.79 | 0.17 | 409 | 84.15 | 2.02 |
| 110 | 82.95 | 1.14 | 210 | 79.92 | 2.80 | 310 | 80.67 | 0.93 | 410 | 85.15 | 1.22 |
| 111 | 75.20 | 32.44 | 211 | 77.05 | 1.52 | 311 | 81.13 | 4.22 | 411 | 83.62 | 0.38 |
| 112 | 77.36 | 4.11 | 212 | 77.86 | 0.78 | 312 | 87.87 | 1.45 | 412 | 85.02 | 1.81 |
| 113 | 80.42 | 33.75 | 213 | 80.77 | 1.64 | 313 | 80.88 | 2.82 | 413 | 85.07 | 3.56 |
| 114 | 80.03 | 23.20 | 214 | 78.45 | 31.73 | 314 | 78.22 | 3.38 | 414 | 82.60 | 10.01 |
| 115 | 79.57 | 4.06 | 215 | 76.75 | 2.88 | 315 | 78.66 | 1.01 | 415 | 83.07 | 2.01 |
| 116 | 70.28 | 100 | 216 | 73.41 | 55.42 | 316 | 82.30 | 3.13 | 416 | 81.32 | 0.74 |
| 117 | 69.33 | 98.35 | 217 | 70.23 | 35.49 | 317 | 76.20 | 20.28 | 417 | 75.30 | 0.31 |
| 118 | 78.56 | 6.17 | | | | 318 | 83.42 | 7.58 | 418 | 79.61 | 0 |
| 119 | 65.07 | 74.72 | | | | 319 | 81.60 | 1.85 | 419 | 82.23 | 1.80 |
| | | | | | | 320 | 79.19 | 1.29 | | | |

average noise reduction at intervals of 5, 10, 15, and 20 meters is 1.61, 2.83, 3.92, and 5.33 dB, respectively. The sound level is also affected by phenomena such as the Earth's surface. The type of ground between the road (source) and the receiver can have significant impacts on the amount of received sound by the receiver. Different surfaces (vegetation and non-vegetation) absorb sound when it reaches the receiver. The effect of noise barriers on reducing traffic noise at a distance of 5 and 10 meters was 1.61 and 2.83 dB, respectively. Individual trees have a small and zero reduction effect in reducing traffic noise. Parks as sound barriers (with an average width of 15m) resulted in a traffic noise reduction of 2.9 dB. Gardens also reduce the noise received by 4.5 Db. Since most noise reduction occurs within 50 m of the road, the sound level was measured on both sides of the road in the presence and absence of an acoustic barrier. Green spaces within a 50-meter buffer can reduce traffic noise levels by varying amounts, with garden and park areas providing the most significant reduction. On average, gardens, parks, strip trees, single trees, and grass cover were able to reduce traffic noise levels by 4.5, 3.3, 1, 0.3, and 0.1, respectively. These results may be found in other similar studies, but in this study, this issue has been investigated coherently and simultaneously in the form of two approaches. Therefore, the main goal of this paper is to present a new methodology that has not been mentioned in any paper so far, and this work is the first research of its kind that has simultaneously estimated the amount of supply and demand for noise reduction ecosystem service. The results and methodology of this research can be used in similar areas to estimate the supply and demand of noise reduction. Also, decision-makers can take management actions to increase supply and meet demand based on the output maps.

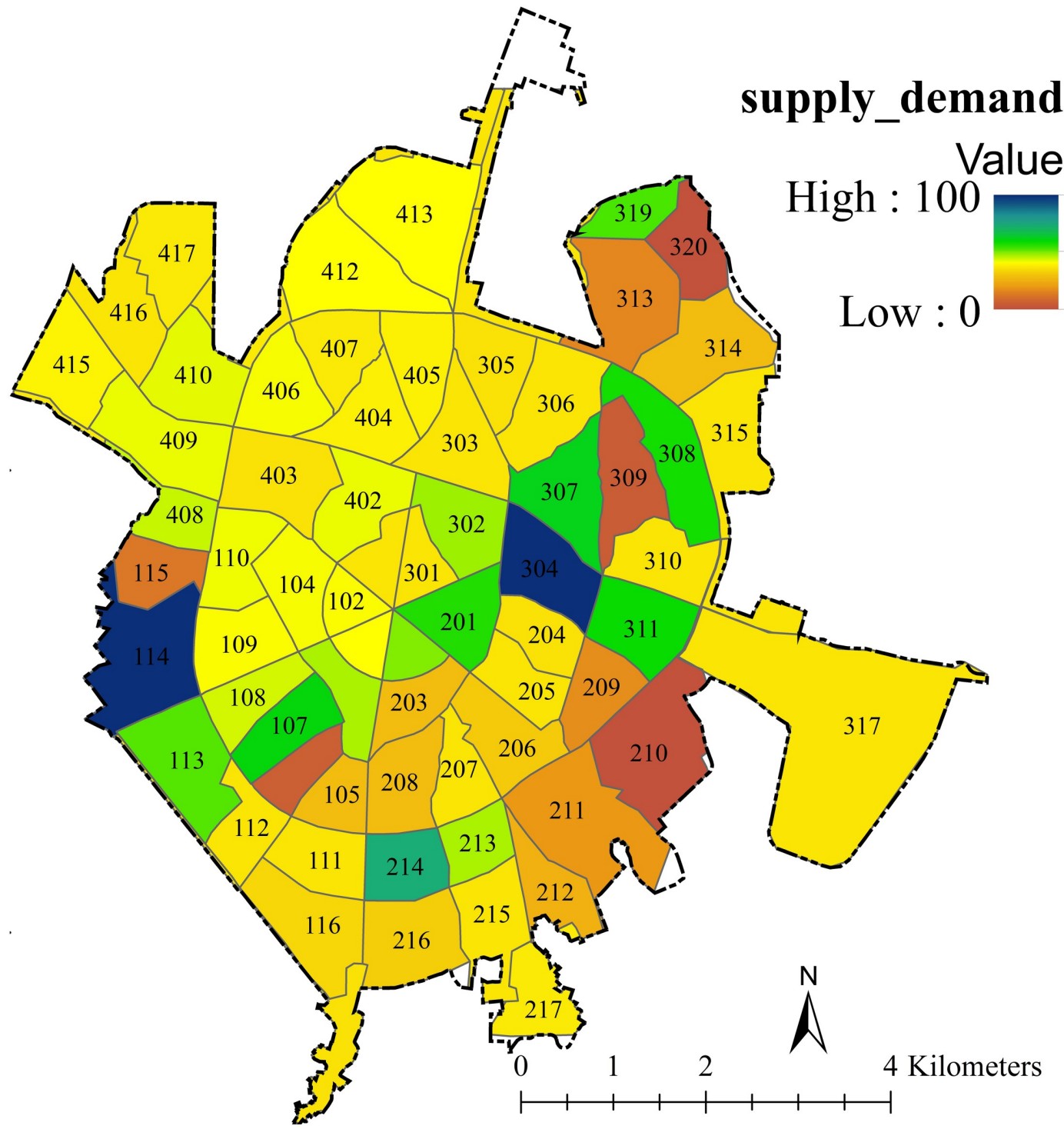

**Fig 8. The supply-demand ratio of noise reduction service based on working units.**

## Supporting information

**S1 File. All used data in this study.**
(XLSX)

## Author Contributions

**Methodology:** Shiva Gharibi.

**Resources:** Shiva Gharibi.

**Software:** Shiva Gharibi.

**Supervision:** Kamran Shayesteh.

**Writing – review & editing:** Kamran Shayesteh.

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
