## [Decision Letter · Decision Letter 0]

31 Oct 2023

PONE-D-23-22927Evaluation of flow, supply, and demand for noise reduction in urban area, Hamadan in IranPLOS ONE

Dear Dr. Gharibi,

Thank you for submitting your manuscript to PLOS ONE. After careful consideration, we feel that it has merit but does not fully meet PLOS ONE’s publication criteria as it currently stands. Therefore, we invite you to submit a revised version of the manuscript that addresses the points raised during the review process.

We look forward to receiving your revised manuscript.

Kind regards,

Chih-Da Wu

Academic Editor

PLOS ONE

“I have read the journal's policy and the authors of this manuscript have the following competing interests:

☒ The authors declare that they have no known competing financial interests or personal relationships that could have appeared to influence the work reported in this paper.”

5. Please amend either the title on the online submission form (via Edit Submission) or the title in the manuscript so that they are identical.

6. We note that Figures 1,2,3,4,5,6,10 and 11 in your submission contain [map/satellite] images which may be copyrighted. All PLOS content is published under the Creative Commons Attribution License (CC BY 4.0), which means that the manuscript, images, and Supporting Information files will be freely available online, and any third party is permitted to access, download, copy, distribute, and use these materials in any way, even commercially, with proper attribution. For these reasons, we cannot publish previously copyrighted maps or satellite images created using proprietary data, such as Google software (Google Maps, Street View, and Earth). For more information, see our copyright guidelines: http://journals.plos.org/plosone/s/licenses-and-copyright.

a. You may seek permission from the original copyright holder of Figures 1,2,3,4,5,6,10 and 11 to publish the content specifically under the CC BY 4.0 license. 

Reviewers' comments:

Reviewer's Responses to Questions

**Comments to the Author**

1. Is the manuscript technically sound, and do the data support the conclusions?

Reviewer #1: Yes

Reviewer #2: Yes

2. Has the statistical analysis been performed appropriately and rigorously? 

Reviewer #1: Yes

Reviewer #2: Yes

3. Have the authors made all data underlying the findings in their manuscript fully available?

Reviewer #1: Yes

Reviewer #2: Yes

4. Is the manuscript presented in an intelligible fashion and written in standard English?

Reviewer #1: Yes

Reviewer #2: Yes

5. Review Comments to the Author

Reviewer #1: The study aims to investigate the match and mismatch of supply and demand for noise reduction in the Hamadan urban area. Here are my recommendations about the paper:

1. The organization of the introduction section is not elaborate. It repeatedly mentions a single concept across various disjointed segments. Authors are required to restructure it in a more coherent manner to effectively convey the intended message.

2. In the introduction, consider adding relevant statistics about the topic and study area. This can help set the context and highlight the research's significance.

3. The paper requires another round of proofreading to significantly enhance some sentences. Few examples are below:

Introduction - L31-32: One million lives healthy years of life are lost annually in Europe due to the effect of exposure to traffic noise.

Introduction - L45-47: In general, ecosystem services are the conditions and processes through which natural ecosystems and the species that make them up, enhance human well-being.

Quantification and mapping of the demand for noise pollution reduction - L195-196: Demand analysis, based on extracting the sound level from the sound flow map and comparing it with the standard sound limits in these uses, can reflect the demand in each land use.

Discussion and Conclusion - L359-360: Therefore, sound measuring stations were selected so that the ground between the sound source and the receiver would paved and asphalted.

4. In the methodology section, it would be beneficial to provide straightforward definitions for 'flow,' 'supply,' and 'demand' concerning noise reduction. Additionally, considering that the methodology contains numerous subsections, it can be challenging for readers to gain a comprehensive understanding. Therefore, it is advisable to consolidate these subsections into a single section for clarity.

5. In the section on quantifying and mapping traffic noise flow, it mentions the use of the Kriging method to classify noise pollution zone maps. However, there is no explanation provided for this method. Please consider adding an explanation, especially for readers who may not be familiar with it.

6. The quantification and mapping of the supply of noise pollution reduction section is somewhat unclear. The explanation for the two approaches is insufficient, making it difficult to follow the text. It is recommended that the authors provide a comprehensive explanation for the first approach and then proceed to explain the second one

7. In the results section, in lines 222-223, the text mentions that "The GIs were divided into six main categories: agricultural lands, gardens, parks, grass and abandoned lands, single trees, as well as street trees." To make this clearer, it would be helpful to explain the unique characteristics that define each category. Furthermore, offering an explanation for the difference between "single trees" and "street trees" would enhance understanding. Additionally, has this classification approach been used in previous studies?

8. In the Results section, lines 223-224, percentages are provided for five categories (amounting to a total of 100 percent). However, what about the sixth category?

9. Results contain no surprises. Have you learned anything that's not in previous, or do you just confirm common knowledge?

10. The discussion and conclusion sections are quite lengthy and lack coherence. They tend to repeat the results from the results section. It would be beneficial to merge them with the results section to create a new section named 'Results and Discussion,' and then have a separate 'Conclusion' section encompassing the paper's summary (results and discussion), limitations, suggestions for future studies, and practical applications.

Reviewer #2: In my opinion, the manuscript demonstrates scientific suitability for publication. I would highly recommend conducting a thorough grammar check for the work. While the content is legible, there are still instances of poor statements that require rectification. It is advisable to include a concluding recommendation in the abstract.

6. PLOS authors have the option to publish the peer review history of their article (what does this mean?). If published, this will include your full peer review and any attached files.

Reviewer #1: No

Reviewer #2: No

---

## [Author Response · Author response to Decision Letter 0]

20 Jan 2024

Manuscript Number: PONE-D-23-22927

Full Title: Evaluation of flow, supply, and demand for noise reduction in urban area, Hamadan in Iran

Dear Academic Editor, Chih-Da Wu

we appreciate the editor and the reviewers for their detailed observations and comments on the manuscript. All changes can be seen in the manuscript. We have fulfilled their comments and suggestions and wish to submit a revised version of the manuscript for further consideration in the journal. Changes in the revised version of the manuscript are highlighted in green. Below, we also provide a point-by-point response explaining how we have addressed each reviewer's and editors’ comments. all tracked changes are visible in red. We hope it will be accepted. Please let me know if there are any changes required.

(supplementary data, maps, and attribute tables are ready and can be available).

Yours sincerely,

Authors

Reviewer #1: 

Major Revision

Comments

1) The organization of the introduction section is not elaborate. It repeatedly mentions a single concept across various disjointed segments. Authors are required to restructure it in a more coherent manner to convey the intended message effectively. Revised 

2) In the introduction, consider adding relevant statistics about the topic and study area. This can help set the context and highlight the research's significance. Because there are no statistics that completely match the topic (supply and demand), therefore, the presentation of statistics has been omitted. But it will be added if necessary.

3) The paper requires another round of proofreading to significantly enhance some sentences. A few examples are below: Grammatically, not only the few mentioned below but the entire text has been revised.

Introduction - L31-32: One million lives and healthy years of life are lost annually in Europe due to the effect of exposure to traffic noise. Revised

Introduction - L45-47: In general, ecosystem services are the conditions and processes through which natural ecosystems and the species that make them up, enhance human well-being. Revised

Quantification and mapping of the demand for noise pollution reduction - L195-196: Demand analysis, based on extracting the sound level from the sound flow map and comparing it with the standard sound limits in these uses, can reflect the demand in each land use. Revised

Discussion and Conclusion - L359-360: Therefore, sound measuring stations were selected so that the ground between the sound source and the receiver would paved and asphalted. Revised

4) In the methodology section, it would be beneficial to provide straightforward definitions for 'flow,' 'supply,' and 'demand' concerning noise reduction. Additionally, considering that the methodology contains numerous subsections, it can be challenging for readers to gain a comprehensive understanding. Therefore, it is advisable to consolidate these subsections into a single section for clarity. Done

5) In the section on quantifying and mapping traffic noise flow, it mentions the use of the Kriging method to classify noise pollution zone maps. However, there is no explanation provided for this method. Please consider adding an explanation, especially for readers who may not be familiar with it. Added 

6) The quantification and mapping of the supply of noise pollution reduction section is somewhat unclear. The explanation for the two approaches is insufficient, making it difficult to follow the text. It is recommended that the authors provide a comprehensive explanation for the first approach and then proceed to explain the second one. Revised

7) In the results section, in lines 222-223, the text mentions that "The GIs were divided into six main categories: agricultural lands, gardens, parks, grass and abandoned lands, single trees, as well as street trees." To make this clearer, it would be helpful to explain the unique characteristics that define each category. Furthermore, offering an explanation for the difference between "single trees" and "street trees" would enhance understanding. Additionally, has this classification approach been used in previous studies? Revised. These 6 groups were divided based on the authors' opinion.

8) In the Results section, lines 223-224, percentages are provided for five categories (amounting to a total of 100 percent). However, what about the sixth category? Revised

9) Results contain no surprises. Have you learned anything that's not in previous, or do you just confirm common knowledge? The results obtained are seen in very few studies and no articles with this detail has investigated the effect of distance and sound barrier on noise reduction at the same time. Also, the purpose of this study was not only to investigate the effect of sound barrier and distance on noise reduction but also to introduce the method of evaluating supply and demand for this type of ecosystem service which is the first study of its kind.

10) The discussion and conclusion sections are quite lengthy and lack coherence. They tend to repeat the results from the results section. It would be beneficial to merge them with the results section to create a new section named 'Results and Discussion,' and then have a separate 'Conclusion' section encompassing the paper's summary (results and discussion), limitations, suggestions for future studies, and practical applications. Revised 

Reviewer #2: 

Major Revision

Comments

In my opinion, the manuscript demonstrates scientific suitability for publication. I would highly recommend conducting a thorough grammar check for the work. While the content is legible, there are still instances of poor statements that require rectification. It is advisable to include a concluding recommendation in the abstract. 

Grammatically, the whole text was revised.

---

## [Decision Letter · Decision Letter 1]

27 Feb 2024

PONE-D-23-22927R1Evaluation of flow, supply, and demand for noise reduction in urban area, Hamadan in IranPLOS ONE

Dear Dr. Gharibi,

Thank you for submitting your manuscript to PLOS ONE. After careful consideration, we feel that it has merit but does not fully meet PLOS ONE’s publication criteria as it currently stands. Therefore, we invite you to submit a revised version of the manuscript that addresses the points raised during the review process.

We look forward to receiving your revised manuscript.

Kind regards,

Chih-Da Wu

Academic Editor

PLOS ONE

Journal Requirements:

Reviewers' comments:

Reviewer's Responses to Questions

**Comments to the Author**

1. If the authors have adequately addressed your comments raised in a previous round of review and you feel that this manuscript is now acceptable for publication, you may indicate that here to bypass the “Comments to the Author” section, enter your conflict of interest statement in the “Confidential to Editor” section, and submit your "Accept" recommendation.

Reviewer #1: (No Response)

2. Is the manuscript technically sound, and do the data support the conclusions?

Reviewer #1: Yes

3. Has the statistical analysis been performed appropriately and rigorously? 

Reviewer #1: Yes

4. Have the authors made all data underlying the findings in their manuscript fully available?

Reviewer #1: Yes

5. Is the manuscript presented in an intelligible fashion and written in standard English?

Reviewer #1: Yes

6. Review Comments to the Author

Reviewer #1: The manuscript has shown significant improvement. However, there are still some areas that require modification. Please find my comments below:

1. Since there are numerous modifications, it is advisable to completely rewrite paragraphs instead of merely crossing out and adding new parts. In the current situation, it becomes challenging for reviewers to navigate through the manuscript.

2. In the introduction section, lines 75-76, the presented statistic appears inaccurate: "At least 1.6 million lives are lost every year in Europe due to traffic noise exposure [1]."

3. In line 191, could you clarify the meaning of "systematically and randomly" in the context of choosing the location?

4. Several minor grammatical errors persist in the text. For instance:

• Line 583: "dstance."

5. The title for Figure 4 requires correction. According to the legend, the traffic noise flow with barrier corresponds to the left-hand side figure. However, the title states: "Sound level without considering the sound barrier effect (left)." Additionally, for the right-side figure, the legend mentions "traffic noise flow without barriers," while the title reads: "Traffic noise flow and locations of hotspots (right)." Please align the titles with the corresponding figures in the legend.

7. PLOS authors have the option to publish the peer review history of their article (what does this mean?). If published, this will include your full peer review and any attached files.

Reviewer #1: No

---

## [Author Response · Author response to Decision Letter 1]

7 Mar 2024

Manuscript Number: PONE-D-23-22927

Full Title: Evaluation of flow, supply, and demand for noise reduction in urban area, Hamadan in Iran

Dear Academic Editor, Chih-Da Wu

we appreciate the editor and the reviewers for their detailed observations and comments on the manuscript. All changes can be seen in the manuscript. We have fulfilled their comments and suggestions and wish to submit a revised version of the manuscript for further consideration in the journal. Changes in the revised version of the manuscript are highlighted in red. Below, we also provide a point-by-point response explaining how we have addressed each reviewer's comments. We hope it will be accepted. Please let me know if there are any changes required.

(supplementary data, maps, and attribute tables are ready and can be available).

Yours sincerely,

Authors

Reviewer #1: The manuscript has shown significant improvement. However, there are still some areas that require modification. Please find my comments below:

1. Since there are numerous modifications, it is advisable to completely rewrite paragraphs instead of merely crossing out and adding new parts. In the current situation, it becomes challenging for reviewers to navigate through the manuscript. All changes have been made in previous version. The final version was attached without track changes.

2. In the introduction section, lines 75-76, the presented statistic appears inaccurate: "At least 1.6 million lives are lost every year in Europe due to traffic noise exposure [1]." revised 

3. In line 191, could you clarify the meaning of "systematically and randomly" in the context of choosing the location? Revised 

4. Several minor grammatical errors persist in the text. For instance.

• Line 583: "dstance." : Edited.

5. The title for Figure 4 requires correction. According to the legend, the traffic noise flow with barrier corresponds to the left-hand side figure. However, the title states: "Sound level without considering the sound barrier effect (left)." Additionally, for the right-side figure, the legend mentions "traffic noise flow without barriers," while the title reads: "Traffic noise flow and locations of hotspots (right)." Please align the titles with the corresponding figures in the legend. Edited.

---

## [Decision Letter · Decision Letter 2]

23 Apr 2024

Evaluation of flow, supply, and demand for noise reduction in urban area, Hamadan in Iran

PONE-D-23-22927R2

Dear Dr. Gharibi,

We’re pleased to inform you that your manuscript has been judged scientifically suitable for publication and will be formally accepted for publication once it meets all outstanding technical requirements.

Kind regards,

Chih-Da Wu

Academic Editor

PLOS ONE

Additional Editor Comments (optional):

Reviewers' comments:

Reviewer's Responses to Questions

**Comments to the Author**

1. If the authors have adequately addressed your comments raised in a previous round of review and you feel that this manuscript is now acceptable for publication, you may indicate that here to bypass the “Comments to the Author” section, enter your conflict of interest statement in the “Confidential to Editor” section, and submit your "Accept" recommendation.

Reviewer #1: (No Response)

2. Is the manuscript technically sound, and do the data support the conclusions?

Reviewer #1: Yes

3. Has the statistical analysis been performed appropriately and rigorously? 

Reviewer #1: Yes

4. Have the authors made all data underlying the findings in their manuscript fully available?

Reviewer #1: Yes

5. Is the manuscript presented in an intelligible fashion and written in standard English?

Reviewer #1: Yes

6. Review Comments to the Author

Reviewer #1: The authors have made necessary improvements; however, the title for Fig. 3 is still incorrect:

Fig. 3 Title:

"Fig 3. Traffic noise flow and locations of hotspots with considering (left); and without considering the sound barrier effect (left) based on the Kriging method"

Both descriptions in the title refer to the left map.

7. PLOS authors have the option to publish the peer review history of their article (what does this mean?). If published, this will include your full peer review and any attached files.

Reviewer #1: No

---

## [Editor Report · Acceptance letter]

10 May 2024

PONE-D-23-22927R2 

PLOS ONE

Dear Dr. Gharibi, 

I'm pleased to inform you that your manuscript has been deemed suitable for publication in PLOS ONE. Congratulations! Your manuscript is now being handed over to our production team.

Kind regards, 

on behalf of

Professor Chih-Da Wu 

Academic Editor

PLOS ONE